# Sertindole, an Antipsychotic Drug, Curbs the STAT3/BCL-xL Axis to Elicit Human Bladder Cancer Cell Apoptosis In Vitro

**DOI:** 10.3390/ijms241411852

**Published:** 2023-07-24

**Authors:** Chao-Yu Hsu, Wei-Ting Yang, Ju-Hwa Lin, Chien-Hsing Lu, Kai-Cheng Hu, Tsuo-Hung Lan, Chia-Che Chang

**Affiliations:** 1Division of Urology, Department of Surgery, Tungs’ Taichung MetroHarbor Hospital, Taichung 435403, Taiwan; t4361@ms.sltung.com.tw; 2Doctoral Program in Translational Medicine, National Chung Hsing University, Taichung 402202, Taiwan; chlu@vghtc.gov.tw; 3Department of Life Sciences, National Chung Hsing University, Taichung 402202, Taiwan; g110052312@mail.nchu.edu.tw (W.-T.Y.); g111052302@mail.nchu.edu.tw (K.-C.H.); 4Department of Biological Science and Technology, China Medical University, Taichung 406040, Taiwan; linjh@mail.cmu.edu.tw; 5Department of Obstetrics and Gynecology, Taichung Veterans General Hospital, Taichung 407219, Taiwan; 6Tsaotun Psychiatric Center, Ministry of Health and Welfare, Nantou 542019, Taiwan; 7School of Medicine, College of Medicine, National Yang Ming Chiao Tung University, Taipei 112304, Taiwan; 8Center for Neuropsychiatric Research, National Health Research Institute, Miaoli 350401, Taiwan; 9Institute of Clinical Medicine, National Yang Ming Chiao Tung University, Taipei 112304, Taiwan; 10Graduate Institute of Biomedical Sciences, Rong Hsing Translational Medicine Research Center, The iEGG and Animal Biotechnology Research Center, National Chung Hsing University, Taichung 402202, Taiwan; 11Department of Medical Laboratory Science and Biotechnology, Asia University, Taichung 413305, Taiwan; 12Department of Medical Research, China Medical University Hospital, Taichung 404327, Taiwan; 13Traditional Herbal Medicine Research Center, Taipei Medical University Hospital, Taipei 110301, Taiwan

**Keywords:** sertindole, antipsychotics, STAT3, BCL-xL, apoptosis, bladder cancer

## Abstract

Bladder cancer is the leading urinary tract malignancy. Epidemiological evidence has linked lower cancer incidence in schizophrenia patients to long-term medication, highlighting the anticancer potential of antipsychotics. Sertindole is an atypical antipsychotic agent with reported anticancer action on breast and gastric cancers. Yet, sertindole’s effect on bladder cancer remains unaddressed. We herein present the first evidence of sertindole’s antiproliferative effect and mechanisms of action on human bladder cancer cells. Sertindole was cytotoxic against bladder cancer cells while less cytotoxic to normal urothelial cells. Apoptosis was a primary cause of sertindole’s cytotoxicity, as the pan-caspase inhibitor z-VAD-fmk rescued cells from sertindole-induced killing. Mechanistically, sertindole inhibited the activation of signal transducer and activator of transcription 3 (STAT3), an oncogenic driver of bladder cancer, as sertindole lowered the levels of tyrosine 705-phosphorylated STAT3 along with that of STAT3′s target gene BCL-xL. Notably, ectopic expression of the dominant-active STAT3 mutant impaired sertindole-induced apoptosis in addition to restoring BCL-xL expression. Moreover, bladder cancer cells overexpressing BCL-xL were refractory to sertindole’s proapoptotic action, arguing that sertindole represses STAT3 to downregulate BCL-xL, culminating in the induction of apoptosis. Overall, the current study indicated sertindole exerts bladder cancer cytotoxicity by provoking apoptosis through targeted inhibition of the antiapoptotic STAT3/BCL-xL signaling axis. These findings implicate the potential to repurpose sertindole as a therapeutic strategy for bladder cancer.

## 1. Introduction

Bladder cancer is a complex and highly recurrent disease. It is the most prevalent malignant tumor of the urinary tract and is among the most common cancers globally [1]. In 2020, 62,100 men and 19,300 women in the United States were diagnosed with bladder cancer out of 893,600 and 912,930 new cancer cases, respectively [2]. Although the age-standardized incidence rates (ASIR) vary significantly across different geographical regions, it is projected to continue rising over the next decade [3]. Smoking is known to increase the risk of bladder cancer by 2.5 times compared with non-smokers [4], and approximately 40% of bladder cancer-related deaths are attributed to tobacco smoke [5]. Aromatic amines have been classified as bladder carcinogens, and the slow acetylation of the key enzyme N-acetyltransferase 2 in aromatic amine metabolism is linked to increased susceptibility to tobacco smoke, polycyclic aromatic hydrocarbons, or other occupational carcinogens [6]. Increased consumption of red or processed meat appears to correlate with bladder cancer development positively [7]. Additionally, the age at diagnosis seems to associate negatively with bladder cancer recurrence and progression [8]. Additional risk factors include male gender, dietary habits, genetics, and arsenic content in water [9,10,11]. Approximately 70–75% of patients present with non-muscle invasive disease (NMIBC), while the remaining 25–30% exhibit tumor cell infiltration into the muscular layer of the bladder wall (MIBC) or distant metastasis [10]. NMIBC patients can undergo transurethral resection of bladder tumor (TURBT) and intravesical chemotherapy in the operating room or after surgery [12]. Regardless of diverse treatment modalities, the prognosis of bladder cancer is still poor.

Signal transducer and activator of transcription (STAT) proteins are a family of transcription factors that participate in various intracellular signaling pathways. This family includes STAT1, STAT2, STAT3, STAT4, STAT5a, STAT5b, and STAT6. Among them, STAT3 has been proven to be critically important for cancer progression [13,14]. Aberrant activation of STAT3, observed in many human cancers, triggers tumor progression by upregulating oncogenic gene expression, thereby promoting tumor aggressiveness [15,16,17,18]. Activation of STAT3 occurs through ligand binding to cell-surface receptors, for example, the interleukin 6 (IL-6) receptor, leading to phosphorylation of STAT3 at the tyrosine 705 residue via kinases such as JAK, c-Src, and c-Abl. Phosphorylated STAT3 further undergoes homodimerization and then translocation into the nucleus to upregulate the transcription of genes involved in cell proliferation, differentiation, apoptosis, angiogenesis, inflammation, and immune responses [18,19,20]. In bladder cancer, persistent activation of STAT3 is vital to sustain cell proliferation, promote survival, facilitate metastasis, and contribute to chemoresistance [20]. Tumor tissues from clinical bladder cancer patients have exhibited significantly elevated expression of phosphorylated STAT3 compared with adjacent normal tissues, further confirming the oncogenic role of STAT3 in bladder cancer [21]. Notably, pharmacological inhibition of STAT3 has been proved to sabotage bladder cancer progression in murine model, therefore highlighting STAT3 as a promising target for the development of bladder cancer therapeutics [22].

Epidemiological evidence has associated a lower cancer incidence in schizophrenia patients with long-term medication, thus arguing a potential anticancer effect of antipsychotics [23,24,25]. Sertindole is an atypical antipsychotic medication and has been used in Europe since 1996 for schizophrenia therapy [26,27]. It is a potent antagonist of dopamine receptor D2, serotonin receptor 5-HT2, and α1 adrenergic receptor [27,28]. Sertindole has been shown to provoke lethal autophagy in neuroblastoma cells in vitro, induce antiproliferation of triple-negative breast cancer (TNBC) cells and gastric cancer cells at the in vitro and in vivo levels, and suppress the breast-to-brain metastasis of TNBC cells in vivo [29,30,31]. These findings support that sertindole holds excellent promise as an anticancer drug and warrants further investigation.

Herein, we present the first evidence supporting the anti-bladder cancer effect of sertindole. We demonstrated that sertindole spares normal human urothelial cells while repressing the viability and clonogenicity of various bladder cancer cells via inducing apoptosis-dependent cell death. Mechanistically, we elucidated that sertindole induces apoptosis through targeted inhibition of the pro-survival STAT3/BCL-xL pathway in bladder cancer cells. Our findings highlight the potential of sertindole as a therapeutic strategy for bladder cancer treatment.

## 2. Results

### 2.1. Sertindole Is Cytotoxic to Various Human Urinary Bladder TCC Cell Lines While Showing Less Cytotoxicity to Normal Human Urothelial Cells

To probe sertindole’s anti-bladder cancer potential, we first conducted MTS assays to evaluate whether sertindole induced in vitro cytotoxicity against a panel of human urinary bladder transitional cell carcinoma (TCC) cell lines, including J82, TCCSUP, and T24. As shown in Figure 1A, sertindole dose-dependently reduced TCC cell viability, with an IC_50_ of 11.43 ± 0.63 μM, 13.43 ± 0.89 μM, 14.79 ± 0.95 μM for J82, TCCSUP, and T24, respectively, upon sertindole treatment for 48 h. In contrast, 48 h treatment with sertindole barely slayed normal human urothelial cells, even at a dosage as high as 20 μM (Figure 1A), arguing that sertindole-induced cytotoxicity was selective against malignant bladder epithelial cells. Furthermore, we performed clonogenicity assays to examine sertindole’s effect on the long-term proliferation of TCC cells. It is noted that sertindole suppressed the colony-forming capacity of all tested TCC cell lines in a dose-dependent manner. Particularly, sertindole at 20 μM dropped the colony-formation levels of J82, TCCSUP, T24 and to 19.58 ± 3.97% (*p* < 0.001), 12.24 ± 0.68% (*p* < 0.001), and 21.92 ± 2.28% (*p* < 0.001) compared with their respective drug-free controls (Figure 1B).

### 2.2. Sertindole-Induced Bladder Cancer Cytotoxicity Depends on Apoptosis Induction

The nature of sertindole-induced bladder cancer cytotoxicity was addressed. In all sertindole-treated bladder TCC cell lines, we observed a marked increase in the levels of the cleaved form of poly (ADP-ribose) polymerase (c-PARP), a canonical biochemical hallmark of apoptosis [32] (Figure 2A). Additionally, sertindole-elicited apoptosis was substantiated by the cell-surface exposure of phosphatidylserine, another well-established apoptosis marker, as evidenced by the elevation in the levels of cells bound by Annexin V (Figure 2B). We next investigated the role of apoptosis induction in sertindole-elicited bladder cancer cytotoxicity. To this end, human bladder TCC cell lines were pre-treated with the pan-caspase inhibitor z-VAD-fmk to block the initiation of apoptosis, followed by 24 h-treatment with sertindole and then subjected to Annexin V/propidium iodide dual staining assays and clonogenicity assays. Notably, both sertindole-evoked apoptosis induction and clonogenicity inhibition in human bladder TCC cells were dramatically abolished by pre-treatment with z-VAD-fmk. In particular, z-VAD-fmk pre-treatment lowered the levels of apoptotic populations from 48.88 ± 3.18% to 29.46 ± 2.33% (*p* < 0.001), 31.83 ± 0.98% to 16.18 ± 1.13% (*p* < 0.001), and 42.14 ± 1.87% to 12.67 ± 1.99% (*p* < 0.001) in J82, TCCSUP, and T24 cells treated with 20 μM of sertindole, respectively (Figure 2C).

### 2.3. Suppression of STAT3 Activation Is Pivotal to Sertindole-Induced Human Bladder Cancer Apoptosis

Considering apoptosis is indispensable to sertindole’s cytotoxic action, we then asked how sertindole provokes apoptosis in human bladder TCC cells. Given STAT3’s well-established prosurvival action and fundamental role in promoting bladder cancer genesis and progression [20], we examined sertindole’s effect on STAT3 activity in bladder TCC cells. It is noteworthy that, in all sertindole-treated TCC cell lines, the levels of tyrosine 705-phosphorylated STAT3 (p-STAT3) were lessened dose-dependently, along with a paralleled decrease in the expression levels of BCL-xL, an antiapoptotic protein and a well-established STAT3′s transcriptional target (Figure 3A, top panel). Kinetic analyses further indicated a marked decline in p-STAT3 levels after 12 h-treatment with sertindole (Figure 3A, top panel). To further prove sertindole as an inhibitor of the STAT3 signaling, we evaluated sertindole’s effect on interleukin 6 (IL-6)-induced STAT3 activation. As expected, IL-6 (100 ng/mL) stimulated a marked increase in p-STAT3 levels in all TCC cell lines, whereas sertindole co-treatment abrogated IL-6-upregulated p-STAT3 (Figure 3B).

Given the data above revealed sertindole as an inhibitor of STAT3 activation, either inherent in cells or induced by IL-6, we then asked whether sertindole-induced bladder cancer cytotoxicity was compromised if STAT3 remains active in sertindole-treated cells. Along this line, we generated T24 and TCCSUP clones stably expressing a dominant-active STAT3 mutant, STAT3-C [33], to sustain STAT3 activity in sertindole-treated cells. As shown in Figure 3C, the vector control clones were sensitive to sertindole-evoked apoptosis, as evidenced by the increase in the levels of c-PARP and Annexin V-positive populations; in contrast, sertindole failed to trigger effective apoptosis in STAT3-C stable clones. Likewise, the colony-forming capacity of STAT3-C stable clones was refractory to sertindole’s inhibitory action (Figure 3D).

### 2.4. BCL-xL Downregulation Is Responsible for Human Bladder Cell Apoptosis Resulting from Sertindole-Induced STAT3 Inhibition

It is well-established that the expression of BCL-xL, a potent antiapoptotic member of the BCL-2 protein family, is transcriptionally controlled by STAT3 [34]. Along this line, we observed a marked BCL-xL downregulation along with a decrease in p-STAT3 levels in all sertindole-treated cell lines; in contrast, BCL-xL levels were rescued from sertindole-induced downregulation in cells with ectopic STAT3-C expression (Figure 3C). We further evaluated if BCL-xL downregulation accounted for sertindole’s proapoptotic action on human bladder TCC cells by testing the extent of sertindole-evoked apoptosis in cells without or with stable expression of BCL-xL. As shown in Figure 4A, sertindole downregulated BCL-xL but also triggered apoptosis in TCCSUP and T24 control clones, as evidenced by the elevation in the levels of c-PARP and Annexin V-positive cell populations. Conversely, sertindole-induced apoptosis was impaired in BCL-xL stable clones (Figure 4A), further leading to the rescue of colony-forming capacity (Figure 4B).

### 2.5. Neither JAK2 nor SRC Appears to Involve in Sertindole-Inhibited STAT3 Activation

With the event downstream of STAT3 blockage by sertindole elucidated, we then focused on the event upstream of STAT3 to explore how sertindole blocks STAT3 activation. It is well-recognized that STAT3 can be activated via tyrosine 705 phosphorylation by upstream non-receptor tyrosine kinases Janus kinase (JAK) or SRC in response to stimulation with cytokines or growth factors [35]. Because of that, we tested whether sertindole blocks STAT3 activation by repressing JAK or SRC. As shown in Figure 5A, it seems that sertindole induced JAK2 activation in T24 cells, as revealed by an increase in the levels of JAK2 with dual phosphorylation at tyrosine residues 1007 and 1008 (p-JAK2), while no immunoblot signal of p-JAK2 was detected in sertindole-treated J82 and TCCSUP cells. As to sertindole’s effect on SRC, it appeared that sertindole promoted SRC activation in all tested bladder TCC cell lines, as judged by the upregulation of the active SRC (i.e., tyrosine 406-phosphorylated SRC (p-SRC)) after sertindole treatment (Figure 5B).

## 3. Discussion

We herein present the first report regarding sertindole’s proapoptotic and cytotoxic effects on human bladder TCC cells with the underlying mechanisms of action. Specifically, we started by showing that sertindole is cytotoxic to a panel of human bladder TCC cell lines while showing less cytotoxicity to normal human urothelial cells (Figure 1). Then, we verified that sertindole’s cytotoxic action primarily relies on the induction of apoptosis (Figure 2). Next, mechanistic analyses identified sertindole as an inhibitor of STAT3 activation and highlighted STAT3 blockade as a fundamental mechanism of action underlying sertindole’s proapoptotic and cytotoxic effects on human bladder TCC cells (Figure 3). Furthermore, we revealed that BCL-xL downregulation is a pivotal mechanism downstream of STAT3 blockade to mediate sertindole-induced apoptosis (Figure 4). Lastly, our experimental results implicated that neither JAK2 nor SRC is required for sertindole to block STAT3 activation (Figure 5). To our best knowledge, the aforementioned findings have never been documented previously.

In recent years, drug repurposing has been appreciated as a promising strategy to expedite the progress of drug discovery for various diseases, including cancer [36,37,38]. Accumulating lines of evidence have underscored the potential of repurposing antipsychotics as cancer therapeutics [39]. Particularly, based on their advantage to pass through the blood-brain barrier (BBB), typical antipsychotic drugs such as Chlorpromazine, Fluspirilene, Haloperidol, Penfluridol, Pimozide, and Trifluoperazine have been proven effective against glioblastoma (GBM) [40,41,42,43,44,45]. Besides brain tumors, the antitumor effects of these antipsychotics have been demonstrated on assorted types of malignancies, including breast cancer, colorectal cancer, liver cancer, lung cancer, oral cancer, pancreatic cancer, and prostate cancer [39] and the references therein. Regarding sertindole-induced anticancer effect, it has been documented that sertindole provokes lethal autophagy in human neuroblastoma cells [29], retards breast-to-brain metastasis of TNBC cells [30] and exerts an antiproliferative effect on human gastric cancer cells [31]. Accordingly, for repurposing sertindole as an anticancer drug, future studies are warranted to further explore the anticancer potential and underlying mechanisms of sertindole on a broad range of human malignancies along with in vivo validation in animal tumor models. It should also be noted that certain undesirable effects on cardiovascular function are associated with using sertindole [46]. Thus, applying optimal dosage and carefully monitoring the patient’s cardiovascular function are mandatory to repurpose sertindole for cancer therapy in the future.

Our current study revealed apoptosis as a primary basis of bladder cancer cell death after sertindole treatment, as supported by the z-VAD-fmk-mediated rescue of cells from sertindole-induced killing (Figure 2). This finding is consistent with the in vitro evidence reported in Zhang et al., where apoptosis was identified as the cause of cell death in sertindole-treated SUM159 cells [30]. Still, the observation that z-VAD-fmk failed to restore the viability of sertindole-treated bladder cancer cells completely implicates the possible involvement of additional cell-death mechanisms. It is noted that previous studies have linked autophagic cell death to the antiproliferative effect of sertindole on neuroblastoma cells [29] and TNBC cells [30]. Thus, although not examined here, the role of autophagy in sertindole-induced bladder cancer cytotoxicity is worth elucidating in the future.

Data presented here revealed sertindole as an inhibitor of STAT3 activation but also confirmed STAT3 blockage as an essential mechanism responsible for sertindole-induced apoptosis in bladder cancer cells (Figure 3). A similar finding about sertindole’s inhibitory effect on STAT3 signaling was reported by Dai et al. in the context of human gastric cancer cells [31]. However, this report did not present direct evidence to validate whether STAT3 inhibition is functionally significant to sertindole-evoked anti-gastric cancer action. Still, considering our findings in bladder cancer and theirs in gastric cancer, it is plausible to postulate that blockade of the STAT3 signaling pathway seems to be a general mechanism of sertindole’s anticancer action. Given STAT3 represents a promising target for cancer therapeutics [15,19], the likelihood that sertindole generally targets the STAT3 signaling axis in cancer cells confers a promising translation potential to sertindole as an anticancer agent for the treatment of various human malignancies, including bladder cancer.

BCL-xL is a potent antiapoptotic BCL-2 family protein and a transcriptional target of STAT3 [34,47]. Our mechanistic inquiry identified BCL-xL downregulation as a critical mediator downstream of STAT3 inhibition for sertindole-induced bladder cancer cell apoptosis, as BCL-xL overexpression protected cells from sertindole’s proapoptotic action (Figure 4). Notably, high BCL-xL levels have been proven to be a poor prognostic marker for urothelial carcinoma [48]. Moreover, genetic depletion or pharmacological inhibition of BCL-xL led to increased apoptotic death and enhanced cisplatin sensitization of bladder cancer cells [48,49,50]. Given BCL-xL’s clinical implications in bladder cancer, our finding about sertindole-induced BCL-xL downregulation further underscores the translational potential of sertindole as anti-bladder cancer therapeutics.

Although we established sertindole as an inhibitor of STAT3 activation, either active constitutively or induced by IL-6, how sertindole blocks STAT3 activation remains elusive. STAT3 is known to be activated via tyrosine 705 phosphorylation by its upstream kinases, JAK2 or SRC [35]. Dai et al. reported sertindole likely inhibits JAK2 to suppress STAT3 activation in the context of gastric cancer cells [31]. In current study, however, sertindole appeared to trigger the activation of JAK2 and SRC, thus ruling out the involvement of JAK2 or SRC in STAT3 blockade by sertindole (Figure 5). It is well-recognized that, besides the upstream activator kinases, the levels of tyrosine 705 phosphorylation of STAT3 are also subjected to regulation by a family of protein tyrosine phosphatases, including SHP1 and SHP2 [51,52,53]. Thus, future studies should assess whether sertindole engages these tyrosine phosphatases to block STAT3 activation. It is also noteworthy that Zhang et al. proclaimed that sertindole induces apoptosis in TNBC cells by targeting its cell-surface receptor 5-HT6 [30]. Accordingly, it would be interesting to evaluate whether pharmacological inhibition of 5-HT6 leads to the blockade of STAT3 signaling and the resultant induction of apoptosis in human bladder cancer cells.

The present study certainly unraveled the cytotoxic effect of sertindole on bladder cancer cells and the blockade of STAT3 signaling as an integral mechanism of action in this process. Still, it should be noted that these findings were derived solely at the in vitro level. Accordingly, the in vivo evidence of sertindole to retard bladder tumor growth and lower tyrosine 705-phosphorylated STAT3 levels within bladder tumors is indispensable to substantiating sertindole’s potential as an option for treating bladder cancer. The in vivo validation of sertindole’s anti-bladder cancer effect will be a primary goal to achieve in our laboratory.

In conclusion, we herein unravel the potential anti-bladder cancer effect of sertindole for the first time. Sertindole slays malignant urothelial cells by impeding STAT3/BCL-xL signaling axis to provoke apoptotic death. Collectively, the current evidence supports the translation potential of sertindole as an anticancer agent for bladder cancer therapy.

## 4. Materials and Methods

### 4.1. Chemicals

Both sertindole and z-VAD-fmk were purchased from Cayman Chemical (Ann Arbor, MI, USA). Recombinant human IL-6 was acquired from PeproTech (Rehovot, ISR). All chemicals used for cell culture were acquired from Gibco Life Technologies (Carlsbad, CA, USA). Crystal violet, methanol, phosphate buffered saline (PBS), polybrene, and puromycin were all purchased from Sigma-Aldrich (St. Louis, MO, USA).

### 4.2. Plasmids

pBabe-HA-STAT3-C, the plasmid encoding the N-terminal hemagglutinin (HA)-tagged dominant-active STAT3 mutant (STAT3 (A661C/N663C); STAT3-C), was previously described [54]. To construct the vector for ectopically expressing BCL-xL, the open reading frame (ORF) of human BCL-xL was PCR-amplified using HCT116 cells-derived cDNA pools as the template by the following primer pair: 5′-ACCGGTGGACTGGTTGAGCCCATCC-3′ (forward) and 5′-CTCGAGGTCAGTGTCTGGTCATTTC-3′ (reverse). The PCR-amplified DNA fragment was TA-cloned into the pCR^®^II-TOPO^®^ vector (ThermoFischer Scientific, Waltham, MA, USA) for ensuing sequence validation. The cloned BCL-xL ORF was then digested by Age I and Eco RI, followed by directional subcloning to the pBabe-HA vector established in our laboratory to generate the expression vector pBabe-HA-BCL-xL.

### 4.3. Cell Culture

Normal human urothelial cells were obtained from ScienCell Research Laboratories (Carlsbad, CA, USA) and cultured in Urothelial Cell medium as recommended by the manufacture. Human urinary bladder transitional cell carcinoma (TCC) cell lines J82 (ATCC HTB-1™) and TCCSUP (ATCC HTB-5™) were purchased from American Type Culture Collection (ATCC) (Manassas, VA, USA) and were grown in Eagle’s Minimum Essential Medium. Human bladder TCC cell line T24 (ATCC HTB-4™) was acquired from Bioresource Collection and Research Center (BCRC) (Hsinchu, Taiwan) and cultured in McCoy’s 5a medium. All culture media were supplemented with non-essential amino acids, 1 mM sodium pyruvate, 10% fetal bovine serum, and 1% penicillin–streptomycin. Cells were grown at 37 °C in a humidified environment with 5% CO_2_.

### 4.4. Cytotoxicity Assay

Sertindole’s cytotoxic effect on normal urothelial cells and bladder TCC cell lines was evaluated based on short-term in vitro cytotoxicity using the CellTiter 96^®^ AQueous One Solution Cell Proliferation Assay (MTS) assay (Promega; Madison, WI, USA) and long-term antiproliferatve activity via clonogenicity assay as documented previously [54,55,56]. For MTS assay, J82, TCCSUP and T24 were cultured in a 96-well plate at a density of 7 × 10^3^ cells/well for 24 h prior to drug treatment. Afterward, the cells were treated with sertindole for 24 h and 48 h, followed by 2 h incubation with 100 μL of MTS reagent and subsequent measurement of absorbance at 490 nm using Sunrise^TM^ absorbance reader (Tecan, AUT). Results are presented as a percentage of the drug-free controls. For evaluating clonogenicity, 4 × 10^5^ cells were first subjected to 24 h-treatment with sertindole (0, 10, 20 μM), followed by seeding sertindole-treated cells (2 × 10^2^) onto 6-well plates to grow in drug-free culture media for 10~14 days to form colonies. Colonies were then washed with 1X PBS, fixed for 15 min in 100% of methanol, and then washed with 1X PBS. Then, colonies were exposed by staining for 30 min with 1% crystal violet solution, rinsed with ddH_2_O, and dried at room temperature (RT) thereafter. Colony numbers in each well was recorded using microscopy.

### 4.5. Immunoblotting

Immunoblotting was executed according to our published protocols [54,55,56]. Briefly, 30 μg of protein lysates from control of sertindole-treated cells were subjected to SDS-PAGE, blot transfer to PVDF membranes, incubation with primary and secondary antibodies, and then exposed to enhanced chemiluminescence (ECL) reagents to reveal protein signals. For IL-6 stimulation, cells were treated with sertindole for 23.5 h and then co-treated with IL-6 for 30 min, followed by lysate preparation and immunoblotting. To block sertindole-induced apoptosis, TCC cells were exposed to z-VAD-fmk (50 μM) for two hours prior to 24 h-treatment with sertindole (0, 10, 20 μM), followed by lysate preparation and immunoblotting. Primary antibodies against BCL-xL (#2764), HA-tag (#3724), cleaved PARP (#9541), phospho-STAT3 (Tyr 705) (#9131), phosphor-SRC (Tyr 416) (#6943), SRC (#2108), and JAK2 (#3230) were acquired from Cell Signaling Technology (Boston, MA, USA). Anti-STAT3 antibody (GTX104616) was obtained from GeneTex (Irvine, CA, USA). Anti-phospho-JAK2 (Tyr 1007/1008) (ab32101) was bought from Abcam (Cambridge, UK). Anti-β-actin antibody (60008-1-1 g) was purchased from Proteintech (Rosemont, IL, USA).

### 4.6. Apoptosis Assay

We evaluated the extents of apoptosis in sertindole-treated bladder TCC cells by the use of Muse^®^ Annexin V & Dead Cell Assay Kit (Millipore; Burlington, MA, USA). Briefly, 3 × 10^5^ cells/well on a 6-well plate were treated for 24 h with sertindole (0, 10, 20 μM) for 24 h. Cells were then resuspended by trypsinization, washed twice with phosphate-buffered saline (PBS), and then incubated with 100 μL of Annexin V & Dead Cell reagent at room temperature in the dark for 20 min. Finally, flow cytometry analysis was performed on the Muse^®^ Cell Analyzer (Millipore; Burlington, MA, USA) according to the manufacture’s instruction to score the levels of Annexin V-positive (apoptotic) cell population.

### 4.7. Establishment of HA-STAT3-C and HA-BCL-xL Stable Clones in TCCSUP and T24 Cells

Cell clones of TCCSUP and T24 cells with stable expression of HA-STAT3-C or HA-BCL-xL were generated by infecting cells with the retroviral particles derived from pBabe-HA-STAT3-C or pBabe-HA-BCL-xL, respectively. To produce pBabe vector-derived retroviral particles, 293 T cells grown to approximately 70–80% confluence were transfected by JetPEI^TM^ (Polyplus-transfection^®^ SA; Illkirch, FRA) with 2.5 μg of the pBabe.puro empty vector, pBabe-HA-STAT3-C, or pBabe-HA-BLC-xL, along with the plasmids expressing gag-pol (2.5 μg) and VSVG proteins (0.25 μg) required for viral particle packaging. The culture media were collected 24 h and 48 h post-transfection for centrifugation (11,000 ×g for 3 min at 4 °C) to collect the supernatant for viral particle harvest. Then, TCCSUP and T24 cells were incubated with the viral particle-enriched supernatant replenished with polybrene (8 μg/mL) for 48 h. After infection, cells were subjected to 2 d-treatment with puromycin (2 μg/mL) for positive selection. The ectopic expression of STAT3-C or BCL-xL in the cell clones survived from puromycin treatment was validated by immunoblotting for HA expression.

### 4.8. Statistical Analysis

All data were acquired from three separate experiments and are presented as mean ± standard deviation. Statistical differences between two independent experimental groups were analyzed using unpaired two-tailed *t* tests. *p* < 0.05 was regarded as statistically significant.

## Figures and Tables

**Figure 1 ijms-24-11852-f001:**
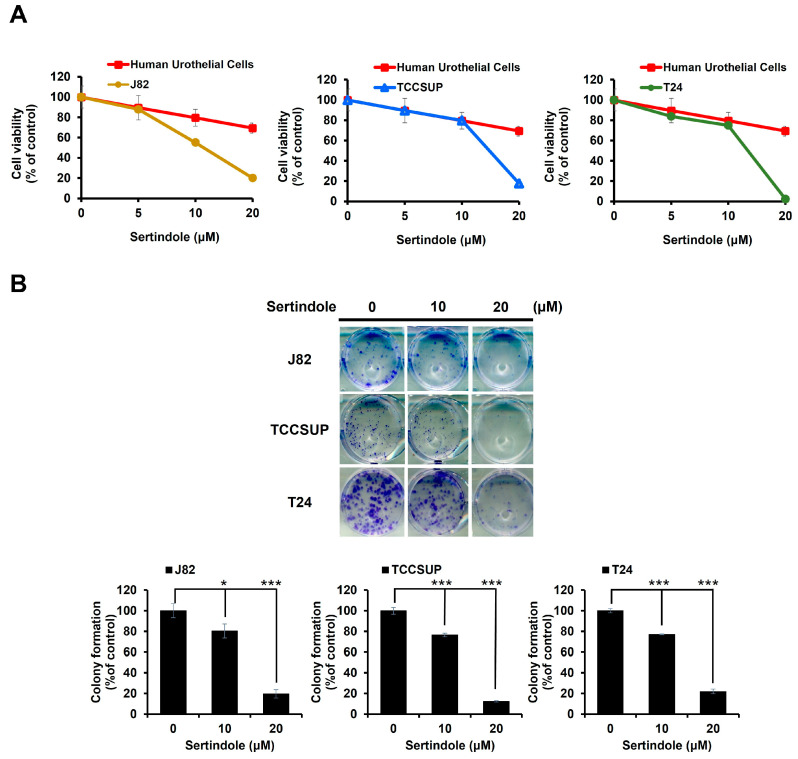
Sertindole is cytotoxic to multiple human bladder TCC cells while sparing normal human urothelial cells. (**A**) MTS viability assay. Sertindole reduces the viability human bladder TCC cell lines J82, TCCSUP, and T24 while showing limited cytotoxicity on normal human urothelial cells up to 20 μM; (**B**) Clonogenicity assay. Sertindole dose-dependently represses the colony-forming capacity of J82, TCCSUP and T24 cells. * *p* < 0.05; *** *p* < 0.001.

**Figure 2 ijms-24-11852-f002:**
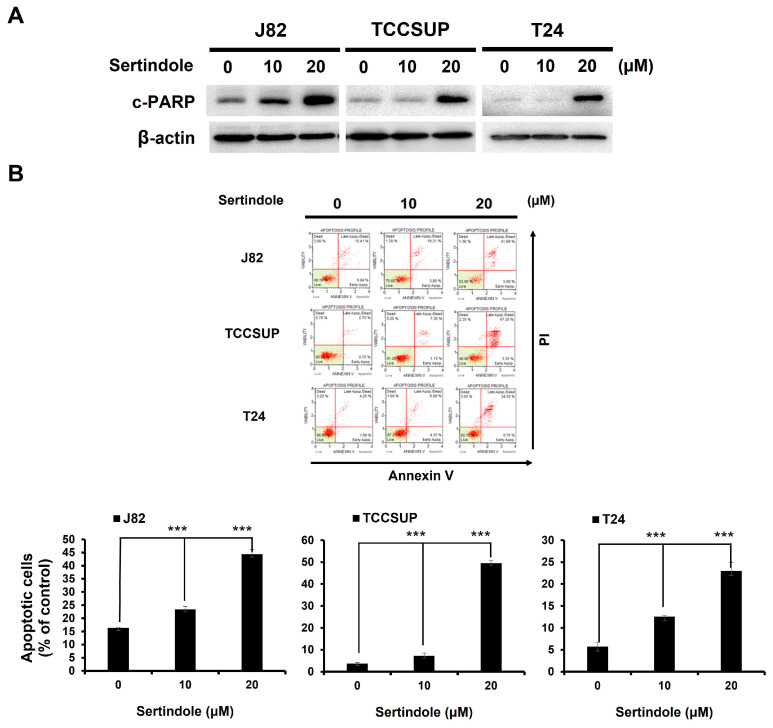
Apoptosis is a main cause of sertindole-induced bladder TCC cell death. (**A**) J82, TCCSUP, and T24 cells were treated with sertindole (0−20 μM) for 24 h, followed by immunoblotting for the levels of PARP cleavage (c-PARP). β-actin was used as the loading control. (**B**) Flow cytometry analysis for the levels of Annexin V-positive (apoptotic) cell populations in sertindole-treated TCC cells. (**C**) Blockade of sertindole-induced apoptosis by the pan-caspase inhibitor z-VAD-fmk (50 μM) rescues bladder TCC cells from sertindole-induced cytotoxicity as gauged by the levels of apoptotic cells and clonogenicity. * *p* < 0.05; ** *p* < 0.01; *** *p* < 0.001.

**Figure 3 ijms-24-11852-f003:**
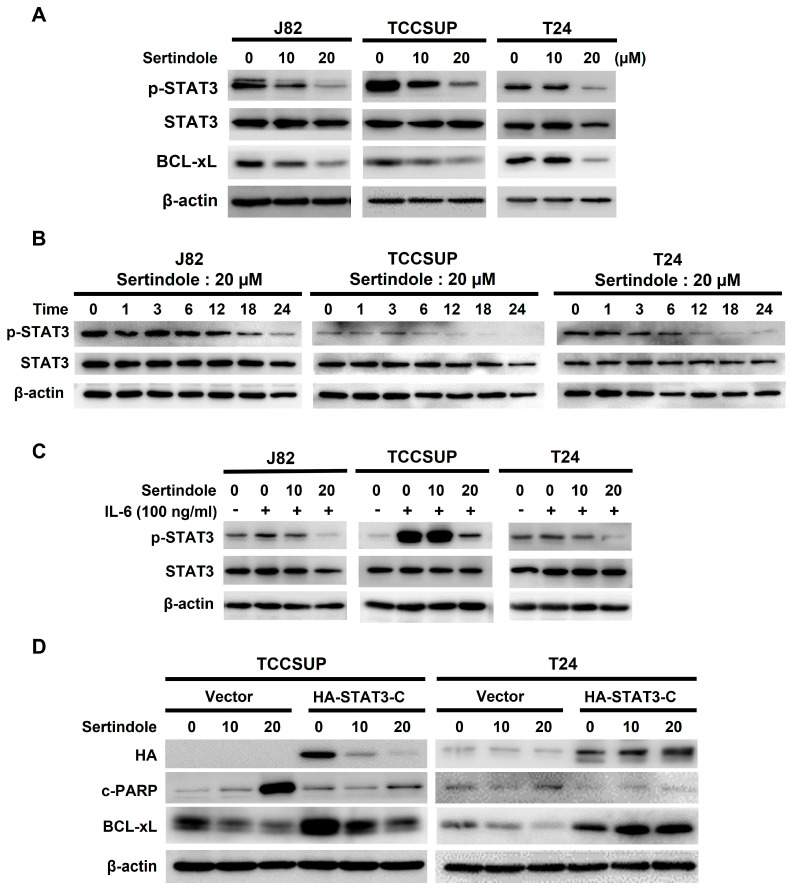
Blockade of the STAT activation is essential for sertindole-induced apoptosis and cytotoxicity in human bladder TCC cells. (**A**) Sertindole induces a dose-dependent reduction in the levels of tyrosine 705-phosphorylated STAT3 (p-STAT3) in the human bladder TCC cell lines J82, TCCSUP, and T24; (**B**) sertindole lowers p-STAT3 levels in a time-dependent manner; (**C**) sertindole blocks interleukin 6 (IL-6)-induced p-STAT3 upregulation; (**D**) Ectopic expression of dominant-active STAT3 (STAT3-C) antagonizes sertindole-mediated induction of apoptosis and reduction in colony-forming capacity of TCCSUP and T24 cells. The restoration of BCL-xL expression supports that STAT3-C sustains the function of STAT3 in sertindole-treated cells. β-actin was used as the loading control. * *p* < 0.05; *** *p* < 0.001.

**Figure 4 ijms-24-11852-f004:**
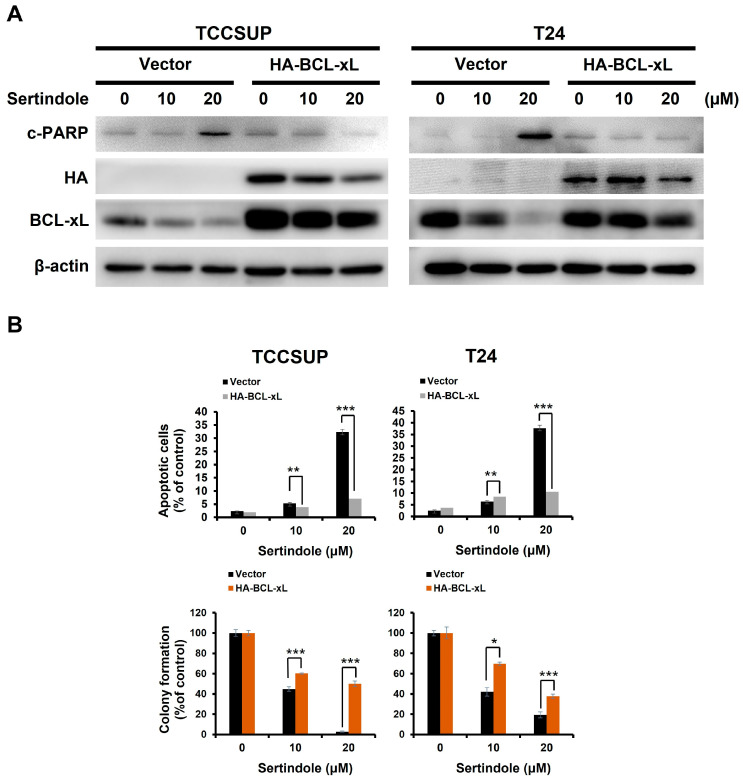
Sertindole downregulates BCL-xL to induce apoptosis/cytotoxicity in human bladder TCC cells. (**A**) Both TCCSUP and T24 cells overexpressing BCL-xL are resistant to sertindole-induced PARP cleavage. β-actin was used as the loading control. (**B**) Ectopic BCL-xL expression abrogates sertindole-mediated apoptosis induction and clonogenicity inhibition. * *p* < 0.05; ** *p* < 0.01; *** *p* < 0.001.

**Figure 5 ijms-24-11852-f005:**
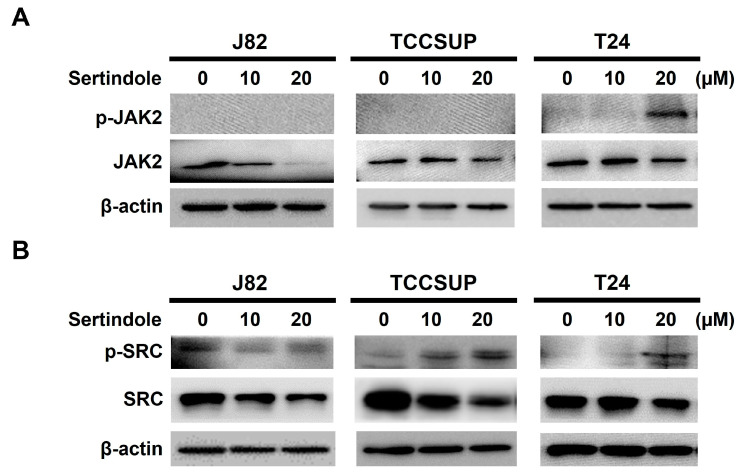
Neither JAK2 nor SRC involves in the inhibitory effect of sertindole on STAT3 activation. J82, TCCSUP, and T24 cells were treated with sertindole (0−20 μM) for 24 h, followed by immunoblotting for the levels of (**A**) total JAK2 and active JAK2 (Tyrosine 1007/1008-phosphorylated JAK2; p-JAK2) and (**B**) the levels of total SRC and active SRC (Tyrosine 416-phosphorylated SRC; p-SRC). β-actin served as the loading control.

## Data Availability

Data will be available by corresponding author (chia_che@dragon.nchu.edu.tw) upon reasonable request.

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
