# Peer review of "Sertindole, an Antipsychotic Drug, Curbs the STAT3/BCL-xL Axis to Elicit Human Bladder Cancer Cell Apoptosis In Vitro"

_ijms, 2023, doi:10.3390/ijms241411852_

Round 1
Reviewer 1 Report
An interesting study on the anticancer mechanism of action of the antipsychotic drug sertindole. The implication of the STAT3 pathway is evidenced with new data, correctly presented and discussed.
The following points should be considered:
1. The difference between cancer cells and normal human urothelial cells (Fig 1) is weak. The anticancer “selectivity” of the product is very modest. Throughout this study, the drug appeared active at 20 uM whereas other studies indicated an action of sertindole on cancer cells at much lower concentrations (<1uM in some cases). The activity level for urothelial cancer cells versus other cancer cell types should be compared (compare published IC50)
2. Sertindole was withdrawn from market due to concern about its cardiovascular toxicity (prolongation of QT intervals, cardiac arrhythmia, some cases of cardiac death). It is therefore very unlikely that this drug can be reintroduced for cancer. A cautionary note should be included in the Discussion.
3. Conclusion Fig 6 is trivial and not very informative. It could be removed.
Author Response
Please refer to the attached file for the authors' responses to the Reviewers' comments. Thank you.

Reviewer 2 Report
Chao et.al., performed experimental analysis of sertindole drug analysis in bladder cancer cells.. It is a good study but for further consideration I have following concerns
Major comments
1) Authors should provide more evidence why they selected sertindole (antipsychotics drug) in comparsion to other antipsycotics related drugs?
2) In result section 2.3 authors used bladder cancer cells to checked sertindole effect on STAT3 activity. Why only STAT3 as there are many other proteins which play role in bladder cancer. It will be good to provide strong premise about STAT3 selection.
3) I suggest authors to provide structural data of sertindole and STAT3 binding site using docking approach. It will help readers to understand more details.
Author Response

(The authors gave the same response as above.)

Reviewer 3 Report
The manuscript entitled “Sertindole, An Antipsychotic Drug, Curbs the STAT3/BCL-xL Signaling Axist to Elicit Human Bladder Cancer Cell Apoptosis” report an interesting findings regarding the potential use of Sertindole in BC, explaining the effects on apoptosis in BC cells. The manuscript is very interesting and could provide interesting points of view as well as future perspectives on future research on the topic. A few corrections are suggested in order to improve the quality of the work and the readability.
TITLE
I would add the fact that is an experimental study in vitro
INTRODUCTION
55-57: concerning the epidemiology of BC, also see: DOI: 10.2144/fsoa-2020-0210.
62: Further data could be provided about the risk factors for BC. See: DOI: 10.3390/cancers14194775 and DOI: 10.1016/j.clgc.2021.12.005
68-70: Regarding the TURBT and the intravesical chemotherapy please be clearer. NMIBC undergo TURBT as a treatment. The intravesical chemotherapy could be done after surgery with MMC, or during the protocol which includes the induction and the maintenance (MMC or BCG according to grading or multifocality). MIBC needs a radical cystectomy. The fact that chemotherapy is a major therapeutic approach is not fully true.
MATERIALS AND METHODS
Provide a scheme reporting the processes leading to your experimentation to increase your findings’ replicability.
RESULTS
116: Check typos
I would suggest you to avoid personal considerations or provide further explanations of the results. Better move considerations, discussion of the findings and future perspectives to the discussion.
DISCUSSION
258: redundant with the introduction.
277: Report if the studies cited with Sertindole were in vitro and if other ongoing trials (both in vitro and in vivo) are coming. Report also the findings of studies in vivo, where available, even in other cancers.
321: Before the conclusions, add the limitations of the study.
Check typos
Author Response

(The authors gave the same response as above.)

Round 2
Reviewer 2 Report
authors answered all my queries. I recommend for publication
Reviewer 3 Report
The authors improved the manuscript accordingly to previous suggestion. No further issues are reported.